# Towards Routine Implementation of Liquid Biopsies in Cancer Management: It Is Always Too Early, until Suddenly It Is Too Late

**DOI:** 10.3390/diagnostics11010103

**Published:** 2021-01-11

**Authors:** Maarten J. IJzerman, Jasper de Boer, Arun Azad, Koen Degeling, Joel Geoghegan, Chelsee Hewitt, Frédéric Hollande, Belinda Lee, Yat Ho To, Richard W. Tothill, Gavin Wright, Jeanne Tie, Sarah-Jane Dawson

**Affiliations:** 1Faculty of Medicine, Dentistry and Health Sciences, University of Melbourne Centre for Cancer Research, Parkville, VIC 3000, Australia; koen.degeling@unimelb.edu.au (K.D.); frederic.hollande@unimelb.edu.au (F.H.); richard.tothill@unimelb.edu.au (R.W.T.); sarah-jane.dawson@petermac.org (S.-J.D.); 2Centre for Health Policy, Faculty of Medicine, Dentistry and Health Sciences, Melbourne School of Population and Global Health, Parkville, VIC 3053, Australia; 3Department of Medical Oncology, Peter MacCallum Cancer Centre, Parkville, VIC 3000, Australia; arun.azad@petermac.org (A.A.); lee.b@wehi.edu.au (B.L.); richard.to@petermac.org (Y.H.T.); Jeanne.tie@petermac.org (J.T.); 4Victorian Comprehensive Cancer Centre, Parkville, VIC 3050, Australia; jasper.deboer@unimelb.edu.au; 5Illumina Inc., San Diego, CA 92122, USA; jgeoghegan@illumina.com; 6Department of Pathology, Peter MacCallum Cancer Centre, Parkville, VIC 3000, Australia; chelsee.hewitt@petermac.org; 7Division of Personalised Oncology, Walter and Eliza Hall Research Institute, Melbourne, VIC 3052, Australia; 8Department of Medical Oncology, Northern Health, Epping, VIC 3076, Australia; 9Department of Surgery, St. Vincents Hospital, Melbourne, VIC 3065, Australia; gavin.wright@svha.org.au; 10Department of Medical Oncology, Western Health, St. Albans, VIC 3021, Australia

**Keywords:** liquid biopsy, ctDNA, CTC, genomics, cancer, minimal residual disease, health services research, biomarkers, health economics, early detection, cancer surveillance

## Abstract

Blood-based liquid biopsies are considered a new and promising diagnostic and monitoring tool for cancer. As liquid biopsies only require a blood draw, they are non-invasive, potentially more rapid and assumed to be a less costly alternative to genomic analysis of tissue biopsies. A multi-disciplinary workshop (*n* = 98 registrations) was organized to discuss routine implementation of liquid biopsies in cancer management. Real-time polls were used to engage with experts’ about the current evidence of clinical utility and the barriers to implementation of liquid biopsies. Clinical, laboratory and health economics presentations were given to illustrate the opportunities and current levels of evidence, followed by three moderated break-out sessions to discuss applications. The workshop concluded that tumor-informed assays using next-generation sequencing (NGS) or PCR-based genotyping assays will most likely provide better clinical utility than tumor-agnostic assays, yet at a higher cost. For routine application, it will be essential to determine clinical utility, to define the minimum quality standards and performance of testing platforms and to ensure their use is integrated into current clinical workflows including how they complement tissue biopsies and imaging. Early health economic models may help identifying the most viable application of liquid biopsies. Alternative funding models for the translation of complex molecular diagnostics, such as liquid biopsies, may also be explored if clinical utility has been demonstrated and when their use is recommended in multi-disciplinary consensus guidelines.

## 1. Introduction

The term liquid biopsy is commonly used to refer to a class of diagnostic methods that analyze circulating tumor markers in body fluids, including blood, saliva or urine, to improve cancer management. In many cases, this is related to the analysis of circulating tumor DNA (ctDNA) and circulating tumor cells (CTCs); however, it could also include circulating RNA, exosomes, and proteins. These components are shed from both primary tumor and metastatic sites and can be used to extract tumor information. Liquid biopsies were first described in post-mortem blood samples in 1869 by an Australian physician, Thomas Ashworth [1]. Ashworth described the presence of circulating tumor cells (CTCs), cells that are shed in blood by the primary tumor, circulate in the vascular system and can cause distant metastases. Since the discovery of CTCs, the liquid biopsy field has made significant progress. Better detection techniques allow the identification and isolation of CTCs and other markers; advanced genomic sequencing methods are now available and applied, and potentially new clinical applications, such as screening and early detection, have emerged. A critical challenge, however, remains the translation of scientific discovery into routine cancer services. Despite an increase in (clinical) research output and a large number of international scientific meetings, the routine use of liquid biopsies is still limited. This paper sought to identify critical issues and the potential way forward to translate liquid biopsies.

### 1.1. The Clinical Utility of Circulating Tumor Cells (CTCs)

Since their discovery, CTCs have been the subject of ongoing research and development with several clinical studies, demonstrating that the number of cells per 7.5 mL blood can prognosticate cancer outcomes in breast, prostate and colorectal cancer [2,3,4,5,6]. Furthermore, it has also been shown that CTCs respond to systemic treatments and thus can give an indication of whether treatment can be effective [7]. In 2005, the Food and Drug Administration (FDA) in the United States (US) cleared the Cellsearch system, a test based on the enumeration of immunomagnetic enriched cells that express EpCAM and Cytokeratins but lack CD45; it therefore became the first approved liquid biopsy test to provide prognostic information in patients with metastatic breast, prostate or colorectal cancer.

Phase III clinical trials evaluating CTC enumeration with clinical parameters have been published in several cancers including the SWOGS0421, SWOG0500, the STIC CTC and the VISNÚ-1 trial [8,9,10,11]. While the SWOG0421 and VISNÚ-1 studies used CTC counts to identify patients with worse outcomes or to stratify patients for aggressive treatment, SWOG0500 and STIC CTC were the only two trials to date that were actually designed to determine the clinical utility of CTCs. The STIC CTC trial concluded that CTCs may be a reliable biomarker to guide the choice between first-line chemotherapy or endocrine therapy in metastatic breast cancer [10].

In metastatic prostate cancer, observational studies but not clinical trials have provided strong evidence that CTCs may also be used to monitor progression on systemic treatment with the potential to discontinue ineffective treatment earlier [12]. This had led to an expert panel recommending CTCs to be routinely used in clinical management of prostate cancer [13]. However, although CTCs can be isolated, have prognostic value, potentially have predictive value and are clinically recommended, they are not routinely used to support cancer management. Although there are concerns regarding the sensitivity and reliability of detecting CTCs in some cancers such as non-small cell lung cancer (NSCLC) [14], the lack of clinical utility is considered the main barrier for routine clinical use [15].

### 1.2. Beyond CTCs: The Clinical Use of Circulating Tumor DNA (ctDNA)

In recent years, there has been an increased interest in the use of other liquid biopsies to facilitate cancer management [16]. In particular, circulating tumor DNA (ctDNA) is a frequent subject of clinical research and has several advantages over CTCs [17]. While CTCs have the advantage of extracting detailed information at the single cell level, ctDNA has the advantage of being easier to detect and the ability to monitor clonal evolution and tumor heterogeneity, which potentially makes it a better candidate for early detection and for management of progressive disease [18]. Other advantages of ctDNA over CTCs that increase the likelihood of clinical application are the ease of sample collection and the ability to store those at room temperature for up to 7 days if Streck tubes are used. Despite the advantages, there are also challenges with the isolation and extraction of ctDNA at extreme low concentration in plasma impacting the sensitivity of the detection and thus clinical use. The main limitation is the lack of standardization of sample processing and storage and of the techniques for extraction and quantification [19]. A European consensus workshop organized by the International Quality Network Pathology (IQN path) summarized nearly a dozen available blood collection tubes and identified 18 currently available ctDNA extraction methods. The workshop revealed a need for guidelines for standardized procedures and a requirement for an external quality assessment program [20]. In particular, when clinical applications are emerging, these fundamental requirements are to be met to avoid false-negative conclusions. This is particularly important as ctDNA is only found in 75% of patients with metastatic disease or in cancers known for their low ctDNA concentration such as in renal cell, brain or prostate cancers [21]. This low detection rate in some cancers increases the risk of false-negative findings, implying that follow-up tissue biopsies are clinically required in case of a negative test. In addition, for ctDNA to be clinically meaningful, it needs to be able to distinguish ctDNA from cancer and healthy cells. Currently, this is usually be done using mutations known from the primary cancer which will have implications on how ctDNA can be used in the clinical workflow.

Many international studies have been published investigating the clinical applications of ctDNA, for screening, detection of minimal residual disease (MRD) and cancer surveillance. In cancer screening and early detection, one of the first approved blood tests is the Epi ProColon for colorectal cancer screening (FDA approved in December 2016). Since then, several high-impact studies have been published investigating the use of different blood test approaches and artificial intelligence-based classifiers for early detection of up to 50 different cancers [22,23,24].

In 2018, a position statement by the American Society for Clinical Oncology (ASCO) and the College of American Pathologists confirmed that, to date, there is still insufficient evidence of clinical validity and utility for the majority of ctDNA assays [25]. However, evidence on the use of ctDNA assays for treatment selection in advanced cancer is now growing. ctDNA assays have shown high agreement with tumor tissue genotyping, particularly when samples are temporally matched. In lung cancer, evidence of prognostic and predictive value is available for monitoring of the human epidermal growth factor receptor T790M (*EGFR T790M*) resistance to *EGFR* tyrosine kinase inhibitors (*EGFR*-TKI) [26], as reflected by the FDA approval for the Cobas *EGFR* mutation test in 2016 [18].

The use of specific liquid biopsy as a diagnostic is expanding. In 2016, the FDA approved the EpiProColon (approved in December 2016) for colorectal cancer screening. In breast cancer, the plasmaMATCH prospective clinical trial recently showed that ctDNA offers accurate genotyping to enable selection of mutation directed therapies for breast cancer patients [27]. The FDA has approved the Therascreen phosphatidylinositol 4,5-bisphosphate 3-kinase (*PIK3CA)* plasma test for breast cancer (approved in May 2019) to select patients for Alpelisib. Preliminary evidence also suggests that tracking the emergence and decline in rat sarcoma virus oncogene (*RAS*) mutated clones driving resistance in metastatic colorectal cancer patients treated with anti-EGFR antibodies may guide the timing of anti-*EGFR* rechallenge [28,29]. More recently, the Guardant360 and FoundationOne Liquid CDx tests (both in October 2020) were approved, both based on next generation sequencing (NGS) designed to inform treatment decisions.

In addition to molecular profiling, liquid biopsies have also been evaluated for their prognostic validity and, although most clinical studies in the early-stage disease setting are relatively small and retrospective, there is growing evidence that ctDNA has prognostic value. Most studies have evaluated quantitative ctDNA as a prognostic marker for survival following curative treatment or aimed to guide decisions in the (neo-) adjuvant setting in colorectal, breast, lung, bladder and pancreatic cancer and melanoma [30,31,32,33,34,35,36,37,38]. In addition, a few systematic reviews have identified evidence levels for a range of circulating tumor markers, including cells, proteins, DNA and RNA, in lung and breast cancer, respectively [26,39]. For breast cancer, about 70 of the 320 studies reviewed investigated the clinical validity or utility of either CTCs or proteins in prospective studies [39].

### 1.3. Clinical Translation, Cost-Effectiveness and Reimbursement

Successful clinical translation of liquid biopsies, including coverage of the cost of testing, in most jurisdictions requires a value proposition weighing the incremental costs of the assay against the clinical benefit [40]. Only few micro-costing studies have been published so far on liquid biopsies covering the costs of the assay as well as the costs of taking the blood samples and analysis time. Two studies have presented detailed micro-costing of ctDNA-based hotspot panels to detect either *KRAS* or *EGFR* mutations for the treatment of colorectal or lung cancer. They concluded that the costs per assay are about AUD 400 (about USD 320) for a single test, which is close to the current reimbursement levels in Australia using tissue-based tests [41,42].

However, the use of comprehensive gene panels comes at a significant cost which may limit their routine use with estimates of USD 1750 [30] and USD 1500 [43] for a single test. These cost estimates do at least flag the need for a detailed health economic analysis as a requirement for routine implementation. So far, only three early health economic studies have been published evaluating the cost-effectiveness of liquid biopsies. Degeling et al. developed a model evaluating the use of CTCs to determine progressive diseases in mCRPC on systemic treatment and conclude there is a potential to avoid overtreatment and reduce costs to the health system [44]. Kapoor et al. modelled the use of miRNA blood-based testing to screen for gastric cancer and concluded that a relatively inexpensive assay of USD 200 is cost-effective [45]. Finally, Sánchez-Calderón et al. used a comprehensive ctDNA panel to determine treatment resistance in HER2-positive breast cancer and concluded this not to be cost-effective [43]. While the cost-effectiveness of liquid biopsies is a requirement for adoption and reimbursement in many countries with a Health Technology Assessment (HTA) program, a recent study showed an increase in private and public payers covering the costs of liquid biopsy assays [46]. Tests reimbursed in the US include those that received market authorization from the FDA, but there are other tests that have received Local Coverage through Medicare without receiving FDA clearance, examples are Natera’s Signatera test for MRD.

### 1.4. The Challenge of Clinically Translating Liquid Biopsies

On 29 October 2020, a multi-stakeholder invitational workshop was organized with the aim to identify promising new applications of liquid biopsies, identify barriers and discuss opportunities to translate new promising liquid biopsies into the clinical arena for routine application. The workshop was organized around three main applications of liquid biopsies, i.e., early detection, the detection of MRD and cancer surveillance for recurrence [47,48]. In addition, we also discussed the application of liquid-biopsy-based molecular profiling in lung cancer as a substitute for tumor tissue biopsies. This was based on experiences using the Guardant360 liquid biopsy assay in the Non-invasive vs. Invasive Lung Evaluation (NILE) study in lung cancer, demonstrating a higher diagnostic yield and the potential to reduce undertreatment because of an inability to find actionable targets [49].

The workshop included a series of short presentations about current clinical trials, pathology provider perspectives and health economic considerations. Three expert-moderated break-out sessions (early detection, MRD and cancer surveillance) subsequently discussed opportunities to translate liquid biopsies into routine clinical care.

## 2. Materials and Methods

Several reviews have nicely presented the potential advantages of using liquid biopsies in different stages of cancer management [47,48]. In the current workshop, we distinguished three main applications at different stages of disease (early detection, MRD and cancer surveillance) and one application using liquid biopsies for molecular profiling. Participants in the workshop were invited because of their expertise in the application or development of liquid biopsies, particularly ctDNA.

### 2.1. Pre-Workshop Survey

The majority of the workshop participants (*n* = 70) were invited to attend because of their expertise and interest in the field, while another 20 spots were reserved on a first come first serve basis. Finally, 98 people registered for the workshop. All participants and speakers were requested to fill out a questionnaire (*n* = 87 response) before the workshop to learn about their experiences and to prepare the workshop pitches and polls. Six pre-workshop questions were included, of which the results are presented in the supplement.

### 2.2. Workshop Short Presentations

The workshop commenced with six clinical research presentations, two about laboratory and industry perspectives for translating liquid biopsies to molecular pathology labs and two presenting early health economic models evaluating the potential of liquid biopsies. The presentations discussed ongoing clinical trials using ctDNA in early stage pancreatic and colorectal cancer, and metastatic prostate cancer, lung cancer, breast cancer and cancer of unknown primary. All presentations were intended to facilitate a further discussion during the break-out sessions.

### 2.3. Workshop Polls

During the workshop, voluntary and anonymous polls (response *n* = 41) were held to engage participants in the discussion and to determine priorities for further research and collaboration. Two polls were designed as a ranking exercise and presented critical requirements to facilitate implementation of liquid biopsies and priorities to build a compelling health economic evidence base. Participants were asked to rank each of the requirements in order of importance. Ranking data were analyzed at the group level and presented the proportion of respondents selecting the respective requirement as the first or subsequent most important factor.

### 2.4. Break-Out Sessions

Break-out sessions were organized following the presentations allowing 45 min of in-depth discussion around three questions: (1) What is the current evidence?; (2) How can we encourage uptake by clinicians?; (3) What needs to be done to enable routine implementation? Each group was requested to discuss these questions with a specific application in mind to enable a more focused discussion.

The break-out groups were defined for three applications based on the pre-workshop responses:Early detection of cancer, i.e., the use of blood tests for the early detection of (multiple) cancers. This could be the use of pan-cancer blood tests such as GRAIL, Cancer-Seek or PanSEER.MRD, i.e., the use of liquid biopsies following curative treatment to guide adjuvant therapy, such as to guide adjuvant systemic treatment in colorectal or breast cancer.Cancer treatment selection and monitoring, i.e., the use of liquid biopsies to select targeted treatment, monitor progressive disease and response to treatment.

Each break-out session, with about 18 participants each, was moderated by two experts who reported back to the larger group in the final plenary session. Participants were allocated to break-out groups according to their preference and aiming for equal group sizes.

### 2.5. Expert Elicitation of Current Level of Evidence

Following the break-out sessions, a final poll was implemented asking participants to state the current level of evidence for each of the three applications. Five levels of evidence were distinguished and presented on a continuous scale, with (1) technical validity being the ability to detect and quantify a molecular aberration, (2) clinical validity demonstrating the correlation with a clinical outcome such as prognostic value for overall survival, and (3) clinical utility referring to the ability of the liquid biopsy to actually guide treatment decisions that improve clinical outcomes. We finally added (4) “ready to use in clinic” as the level of evidence where clinicians felt the test was ready for use and (5) “cost-effective” being the demonstration of an economically viable test relative to the clinical benefit. An example of a pilot question asking for the evidence for early detection of cancer using a pan-cancer test is presented in Figure 1.

The responses to these questions were used to draw a density plot presenting the probability of evidence available as well as the uncertainty in the estimates by the participants.

## 3. Results

The workshop had a total of 98 registrations and 75 active online participants. During registration for the workshop, 87 participants consented to fill out the questions about their experience with liquid biopsies (Appendix A). Most participants were either clinical or basic researchers, but there was significant representation from industry (*n* = 12), HTA professionals and payers as well as consumer representatives. Most workshop participants had experience using liquid biopsies in lung cancer patients (21%), followed by colorectal cancer, melanoma and breast cancer (each 12%, Appendix A).

The pre-workshop questions showed that more than 50% of the respondents thought the use of liquid biopsies could have most impact for MRD detection and for cancer surveillance, followed by early detection (41%) (Figure 2). Moreover, respondents thought that most scientific progress had been made in MRD and cancer surveillance. This suggests that early detection is considered a promising application but is in a relatively early stage of development. According to the participants, three main barriers are important to navigate in the translation of liquid biopsies, including reimbursement, establishing the clinical utility and standardization of platforms for analysis and for clinical use (Appendix A). While reimbursement is considered the most important barrier, it is conditional on the demonstration of clinical utility to improve clinical management and clinical outcomes. In other words, clinical utility is a requirement for any test to be reimbursed so this must be achieved before reimbursement is even considered.

Additional polling was implemented regarding the role industry and pathology providers could play to ensure successful integration into the clinical and pathology workflows (Figure 3). From this poll (*n* = 41), it was concluded that most (55%) respondents selected the “definition of minimum performance thresholds” to be the most important factor to address.

Setting up quality standards was considered to be the second most important factor by 35% of respondents and third most important by 40% of respondents. Improving workflow and turn-around time was ranked relatively high as a third and fourth criterion. Finally, ensuring availability of ctDNA analysis in all pathology labs, including rural health facilities, was less important assuming sample collection and storage would be possible in regional health centers. Navigation of funding on the Medicare Benefits Schedule (MBS) was also not considered an important factor for industry and pathology providers to take responsibility for.

While few health economic models investigating cost-effectiveness of liquid biopsies are available, some early-stage models have been published. These models synthesize existing data-sources to estimate cost-effectiveness under various clinical scenarios. With sensitivity analysis it is possible to determine the most important drivers for cost-effectiveness, such as the cost or the sensitivity and specificity of the assays. Two simulation models were presented during the workshop to illustrate how these models may be used to determine if the test is economically viable.

Almost 75% of the respondents (*n* = 32 available) thought health economic simulation models were useful to guide implementation into clinical care. Respondents also clearly identified that the establishment of clinical benefits, such as increased survival (60% respondents), is the most important implementation factor to address (Figure 4). Other criteria were almost equally important, with the least important being the number of patients tested and the cost of the liquid biopsy test.

Following the three break-out sessions, participants were asked about the current levels of clinical evidence for the use of liquid biopsies in four different applications.

Figure 5 presents the group averages and distribution for these four applications, with the evidence of *EGFR* T790M resistance monitoring being considered the most developed. This was followed by the use of liquid biopsies for monitoring progressive disease in general and for identifying MRD. The least evidence, with limited uncertainty concluded from the smaller distribution, was found for early detection of cancer. Appendix A showed that clinicians were slightly more positive about the levels of evidence for assays used for early detection and MRD compared to the researchers.

## 4. Discussion

This paper presents the results of a workshop intended to define issues and opportunities for the translation of liquid biopsies into routine health services. While many colleagues recognize demonstrated clinical utility to be a critical step towards routine use, the workshop participants fostered in a highly engaging and constructive discussion about other requirements for using liquid biopsies in our health services. This section discusses the need for tumor-informed assays, the integration in the clinical workflow, optimization of test use, cost of the assays, and finally the opportunities to include assays in clinical guidelines required to inform applications for reimbursement.

### 4.1. Early Detection

While all the respondents concluded that there are opportunities for early detection using liquid biopsies to have impact, the evidence on this application is immature. Several commercial initiatives are underway to improve early detection, with methods focused on either aberrant methylation of ctDNA such as in the GRAIL Galleri test [23] and panSEER [24] or methods based on specific gene mutations and plasma proteins such as in CancerSeek [22]. A test based on methylation patterns of ctDNA may detect a wider range of cancers. For instance, the GRAIL Galleri test is based on targeted methylation sequencing and uses a machine learning classifier to detect more than 50 cancers. CancerSEEK, in contrast, is used to detect eight cancers. The different approaches may have implications for how these tests can be used. Any screening test would require a high specificity as otherwise too many patients would risk being exposed to potentially invasive diagnostic procedures unnecessarily. Current studies have been able to demonstrate the ability to pick up cancers after testing their classifiers in validation cohorts [23]. Although high specificities have been reported, this is based on a comparison of confirmed cancer patients against healthy controls only. New large-scale studies evaluating classifiers in real-life populations, including the UK-SUMMIT (NCT03934866) and the US-PATHFINDER (NCT04241796) studies, are underway and these results are required to determine if these tests add real clinical value. In addition, tests used for screening and early detection should be able to detect cancers at an early stage to make a difference. Comparison of four different classifiers showed that the best test only detects 50% of the stage I lung cancer patients [50]. The experts in the workshop extensively discussed the need to be able to detect the tissue of origin for a screening test and expressed a preference for molecular signatures that are able to do so, such as in a targeted methylation-based assay. From an implementation perspective, this targeted screening is also preferred over pan-cancer screening in a broad population, because the latter can potentially result in sub-optimal or unnecessary follow-up, both leading to unfavorable cost-effectiveness ratios if routine imaging is performed following a positive pan-cancer test. Another concern is that, even though the sensitivity and specificity of a test can be high, the actual positive predictive value of the test is dependent on the prevalence of the cancer in the population to be screened meaning that less common cancers may be missed. This needs to be considered if these tests are to be used for pan-cancer population screening, as the positive predictive value is dependent on the prevalence in the screening population implying detection of some (rare) cancers may be poor. The other question is how these tests should be used, either as an off-the shelf consumer product or supervised under the authority of a professional such as a General Practitioner. The experts raised the issue that there is a strong consumer demand, as also seen in the implementation of non-invasive prenatal testing (NIPT) for newborns, and the willingness to pay for these tests out-of-pocket can drive uptake. Consumers may also accept lower standards for the performance of these tests. The experts in the workshop, however, shared the preference that any screening test for cancer, if validated, be implemented under the guidance of a medical professional, with consideration of baseline risk (germline or polygenic risk scores), diagnostic yield and current screening programs.

### 4.2. Clinical Utility of Liquid Biopsies for MRD, Treatment Selection and Monitoring

In contrast to early detection of cancer, the polls (Figure 4) showed that the experts were in agreement that there is evidence of the prognostic value of liquid biopsies in MRD detection and cancer surveillance and monitoring of metastatic disease. Several studies have demonstrated that it is possible to distinguish patients with poor survival outcomes using quantitative assays, such as the levels of ctDNA [31,38,51]. However, whether clinical management based on these test results improved survival outcomes remains uncertain. According to the experts, “We do know these tests can be done and that they are a strong marker of relapse, but at this point it is hard to say if the detection of MRD has the clinical utility to improve survival or reduce toxicity from treatment”.

The experts preferred a tumor-informed over a tumor agnostic assay for MRD and surveillance, due to the increased ability to identify molecular targets to optimize treatment. However, the increased cost of tumor-informed compared to tumor-agnostic assays indicates the need to consider what the optimal use of these tests is. For MRD, a tumor-informed approach first requires tumor tissue sequencing to find candidate somatic mutations for the design of patient specific targeted gene panels for ctDNA tracking as described in Tie et al. [36] or Yoshinami et al. [52]. Such approaches may be effective for monitoring MRD using ctDNA in tumors with a high mutational burden, such as colorectal cancer, melanoma and pancreas cancer, but may not be as feasible in cancers with fewer DNA mutations, such as prostate and breast cancer. Other approaches for measuring MRD that are currently also being explored, such as analyzing ctDNA methylation, need further validation before they can be used in a clinical setting [53].

A similar challenge of tumor-informed vs. -agnostic assays was identified for monitoring cancer in the metastatic setting, with tumor-informed approaches to be preferred over quantitative assays. Compared to MRD, however, the experts thought that tumor-informed assays may even be more important in the metastatic setting as NGS-based assays can replace solid-tumor biopsies and be used as a companion test to identify particular targets to guide selection of specific treatments. The most obvious example at this moment is the use of NGS-based assays such as Guardant 360 for treatment selection in lung cancer [49]. However, it would create an even greater challenge if this application is considered for serial monitoring of progression and clonal evolution as larger gene panels, costing up to USD 5000, are not economically viable to be performed on a serial basis. The experts therefore also considered alternative testing strategies, such as serial monitoring using a small targeted panel or ddPCR for single markers such as mutations in B-Rapidly Accelerated Sarcoma (*BRAF)*, *EGFR* or *PIK3CA*. A clinically feasible and economically viable approach may be to implement NGS as a first pass and then follow cancer progression using ddPCR, costing only USD 300 per test. It was recognized that the sensitivity of the assays for personalized monitoring should be very high to avoid false negatives, given that these assays may be used to direct the delivery of targeted therapies. However, this needs to be weighed against the risk of high false-positive rates in case ultra-sensitive ctDNA assays are used.

### 4.3. Turn-Around Time

Routine implementation requires ctDNA analysis to be integrated into the clinical workflow. For instance, in tumor-informed assays, candidate mutations for ctDNA tracking need to be first determined using panel-based sequencing on tissue biopsies or tumor specimens obtained from surgery. This potentially increases turn-around time as planning of neo-adjuvant treatment, surgery and adjuvant treatment needs to be aligned in the workflow. Knowing which mutations to test in the plasma before or immediately after surgery, may reduce turn-around-time for ctDNA analysis by a couple of weeks which is a clinically relevant time window. Turn-around time was also mentioned in relation to sequencing volumes. In particular, laboratories receiving few samples for sequencing may wait until more samples can be analyzed simultaneously to make it cost-efficient, thus increasing turn-around times. For the experts, this implies that it is not necessary that each molecular pathology laboratory be able to analyze ctDNA, as long as we ensure ctDNA analysis is available to everyone. A hub-and-spoke model with sites collecting and expert sites analyzing the samples would be an acceptable solution to ensure efficiency and rapid turn-around times to also help patients in regional health centers. This may change if new sequencing methods are available in a few years, as a price drop will increase processing and decrease turn-around time.

### 4.4. Do We Need to Get the Test Right Before Routine Use?

The experts identified challenges to balance the information required to inform clinical management and the costs and turn-around time of the assays. International studies have concluded comprehensive, multi-gene, panels cost between USD 1750 [30] and USD 4000 (https://www.therapyselect.de/en/guardant360/faq). In contrast, hot-spot panels using ddPCR only cost a fraction with about USD 212 for a plasma-based EGFR test [42] and less than USD 100 for KRAS [41].

This leaves important questions to solve. First, in order to improve clinical utility, the experts preferred a tumor-informed test and would recommend waiting for the evidence to be available before considering routine implementation and/or requesting reimbursement. In addition, the costs of NGS will likely drop in the years to come, which may be another argument to wait for the clinical evidence and a price that is more likely to reflect value for money. The downside of this approach is that it may even take longer before the test satisfies our needs as the developments in precision oncology are usually faster than the implementation of new tests. Waiting for the test to be ready before a trial is designed appears relatively inefficient and prone to the risk of always being too late [54]. In particular, in the case of liquid biopsies that are not used as a companion tests, this will be a challenging and timely process as we learned from for instance from Oncotype DX and Mammaprint, which are tests that are currently not reimbursed despite successful completion of several large and rigorous clinical trials. It may thus be recommended to identify promising applications of liquid biopsies early, to fast-track evidence development to demonstrate utility, to promote standardization of test platforms so pathology providers can fast-track a range of assays and to design (adaptive) clinical trials able to include clinical practice changes. Initiatives such as Bloodpac (https://www.bloodpac.org) in the US are critical to collectively build such evidence and to facilitate translation.

### 4.5. Reimbursing Liquid Biopsies and Evidence

It is widely recognized that reimbursement of biomarkers is challenging [55,56]. Most biomarkers reimbursed by healthcare payers (private or public) are co-dependent or companion to a targeted treatment and their reimbursement is part of a bundled test–treatment combination. Moving liquid biopsies as a substitute for an existing companion test, such as for FoundationOne Liquid or Guardant360, may therefore be easier than pushing liquid biopsies that are not directly related to one specific treatment. For instance, *EGFR* monitoring is approved and reimbursed in NSCLC [57]. Learning from these examples may benefit translation of other liquid biopsies. Internationally, there also is a discussion about the evaluation of technologies for reimbursement decisions as the current HTA frameworks are increasingly complex to apply in precision medicine [58]. The diagnostic and treatment landscape is rapidly developing and creating massive challenges for HTA agencies making informed and deliberate decisions about which tests to be funded. It is obviously easy to retrospectively determine which patients, and what stage of their disease, have benefitted from the diagnostic test and treatment. However, it is more difficult to translate this into a reimbursement decision at a population level restricting the coverage to those who are (likely) to benefit. The more complex the detection (which tumor markers and platforms) and analytic methods (NGS, PCR) are, the more difficult it will be to make these decisions. Alternative payment models can be considered instead, such as bundled payment per episode of care, or a lump-sum payment for the diagnostic tests for each patient diagnosed with cancer. This will leave the actual decisions about which test at what stage at the discretion of the multi-disciplinary team and tumor board. If this was a way forward, it is critical that tests are available to everyone and incorporated in clinical guidelines to prevent inequity in access. Therefore, multidisciplinary guideline development for the use of diagnostic tests and liquid biopsies and consensus workshops and reports like the results presented in this paper would be indicated [13,59,60]. These consensus reports ideally are complemented with an early dialogue between consumers, clinicians and payers about evidence that needs to be demonstrated for regulators to make an informed reimbursement decision.

## 5. Conclusions

This workshop was held to discuss if and how emerging developments in the application of liquid biopsies can be translated to routine clinical services. It is concluded that a tumor-informed approach is preferred to be able to demonstrate clinical utility for MRD and cancer surveillance, but that economic arguments may encourage a discussion about optimal test use combining large NGS panels with hot-spot ddPCR methods for different clinical applications. The efficient use of liquid biopsies requires integration in clinical and pathology workflows to reduce turn-around time and maximize impact on clinical decisions. The question of when reimbursement should be sought remains unanswered. However, alternative funding models, such as bundled payments for diagnostic tests, may help with the translation and coverage of new tests for which clinical utility has been demonstrated and therefore are recommended in clinical guidelines.

## Figures and Tables

**Figure 1 diagnostics-11-00103-f001:**
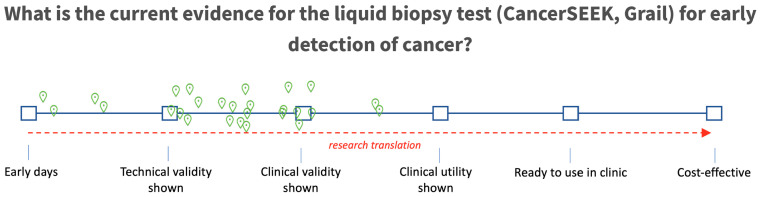
Example of an evidence slider with which the translational research phase was determined. Each workshop participant could vote once (green dots) following a discussion about this case in the break-out sessions. The individual votes are used to construct density plots.

**Figure 2 diagnostics-11-00103-f002:**
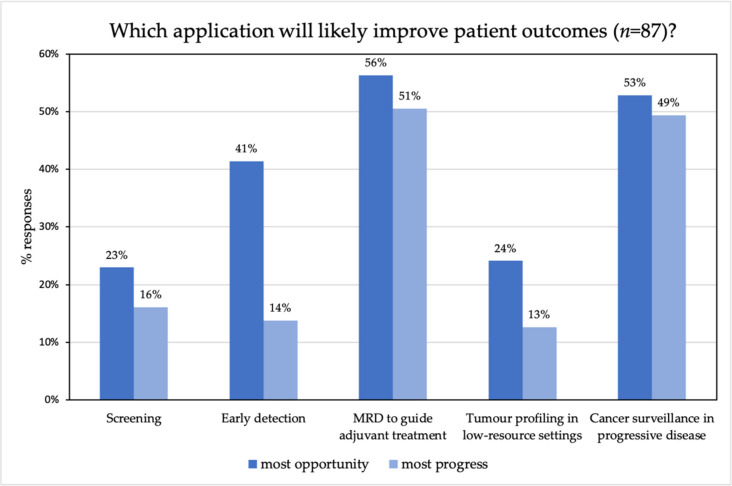
Pre-workshop question asking which application will likely have the most impact and for which application the most scientific progress has been made.

**Figure 3 diagnostics-11-00103-f003:**
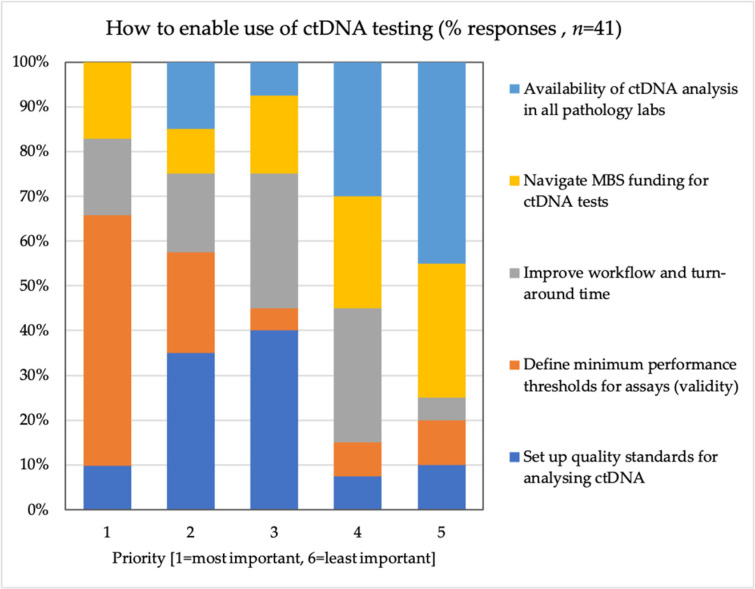
Results of ranking 5 statements related to clinical use of ctDNA.

**Figure 4 diagnostics-11-00103-f004:**
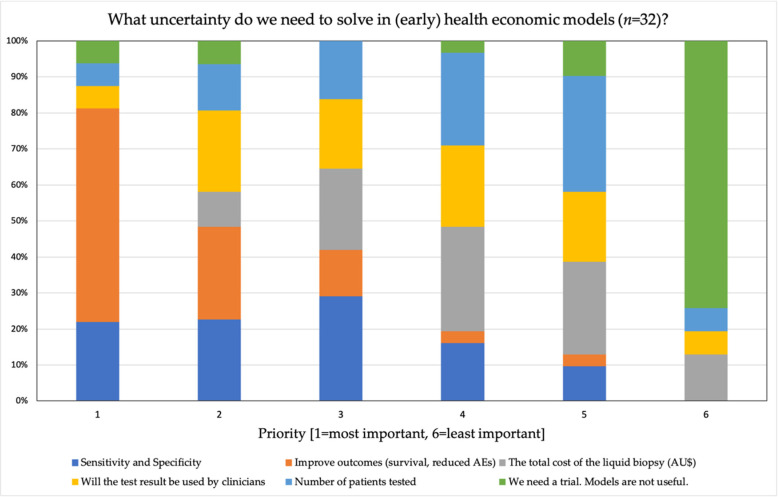
Ranking of statements relevant for health economic modeling of liquid biopsies.

**Figure 5 diagnostics-11-00103-f005:**
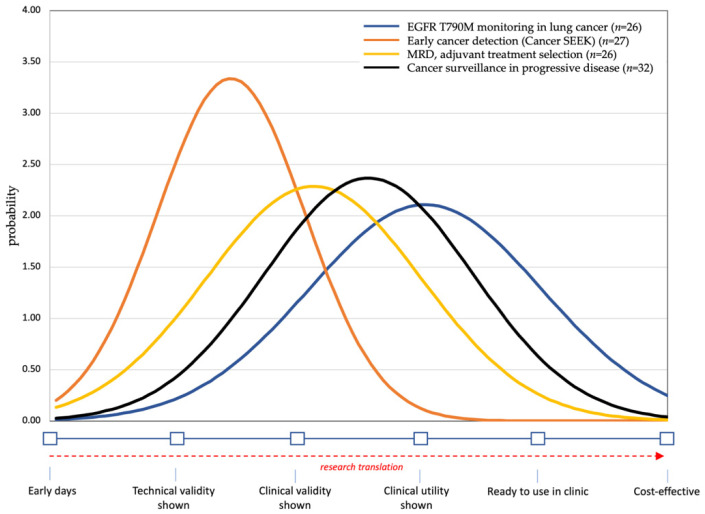
Density plots presenting the translational stage of the 4 ctDNA applications.

## Data Availability

The data presented in this study are available in this article and Appendix A.

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
