# Peer review of "Towards Routine Implementation of Liquid Biopsies in Cancer Management: It Is Always Too Early, until Suddenly It Is Too Late"

_diagnostics, 2021, doi:10.3390/diagnostics11010103_

Round 1
Reviewer 1 Report
The manuscript entitled "Towards Routine Implementation of Liquid Biopsies in Cancer Management: It’s Always too Early, Until Suddenly It’s too Late" highlighted how emerging developments in the application of liquid biopsies can be translated to routine clinical services.
- The Authors should describe in paragraph 1.2 also the most common centrifugation and extraction methods for ctDNA
- The Authors hsould provide the extensive forms for all acronyms, including gene acronyms, through the text when they first appear.
- Gene acronyms should be written in italics.
Author Response
Dear Reviewer,
Thank you reviewing our manuscript and for making suggestions to further improve the report. Please find brief answers to your queries,
Maarten IJzerman
Reviewer 1
The manuscript entitled "Towards Routine Implementation of Liquid Biopsies in Cancer Management: It’s Always too Early, Until Suddenly It’s too Late" highlighted how emerging developments in the application of liquid biopsies can be translated to routine clinical services.
- The Authors should describe in paragraph 1.2 also the most common centrifugation and extraction methods for ctDNA
Re: We agree this improves the quality of the paper and have added a sentence about extraction methods for ctDNA. See lines 100-107 in the revised manuscript (with track changes). We have presented some of the high-level challenges as we don’t think this paper should discuss technical details of different centrifugation and extraction methods with some excellent reviews published elsewhere. We also included two additional references (Deans et al 2019 and Devonshire et al, 2014).
- The Authors hsould provide the extensive forms for all acronyms, including gene acronyms, through the text when they first appear.
- Gene acronyms should be written in italics.
Re: We agree this could have been more consistent and have included the extensive forms and acronyms in italics. Please find more details in the track changes, examples are in lines 127-137 and 444.

Reviewer 2 Report
The article is an original research that deserves the attention of the readers. I would like to note a few points that, in my opinion, should be emphasized:
- Cell-free circulating tumor DNA is found in only about 75% of patients with metastatic tumor disease, with significant differences between different tumor forms and tumor stages. The concentration of cDNA is especially low in renal cell carcinomas, prostate and thyroid carcinomas, as well as in brain tumors (due to the blood-brain barrier), even with metastasis, so it is not always possible to diagnose it.
- Until now, there is no marker that could reliably distinguish ctDNA from tumor cells from cfDNA from healthy cells. Only the detection of a mutation known from the primary tumor gives confidence that the study worked, and the test result has clinical significance. However, if no known mutation is found in the liquid biopsy, then either there is no ctDNA in the blood, or the sensitivity of the assay is too low to detect existing cDNA. Also problematic in this context is the potential release of mutated DNA from benign, inflammatory layering lesions (eg, melanocytic skin nevus, intestinal polyp).
- Detection of ctDNA in blood does not allow drawing conclusions about where the tumor is located and which organ is affected. This should be further explored using imaging diagnostics.
- Liquid biopsies have a high positive but low negative predictive value. For example, detection of an EGFR mutation from blood can be used as a prognostic marker to predict the response to targeted therapy with tyrosine kinase inhibitors, but in the absence of an EGFR mutation in the blood, the tissue sample must be further studied. Cases in which the EGFR mutation is not detected on liquid biopsy, but is detected on biopsy of tumor tissue are relatively common, and therefore, in the absence of signs of mutation in the blood, confirmation of its presence in tumor tissue should be sought.
- An interesting and promising field of application of liquid biopsy is the identification of emerging resistance mutations during tumor progression after therapy in order to adapt further treatment. Liquid biopsy plays an important role in finding the p.T790M resistance mutation in the EGFR gene in targeted therapy for non-small cell lung cancer. An additional molecular test for detecting this mutation of resistance to p.T790M-EGFR can be performed quickly and reliably based on ctDNA from blood plasma. However, the sensitivity of this method is only about 60-70%, therefore, in this situation, in the absence of evidence of mutation of resistance to EGFR, a second biopsy or sampling of cytological material from a progressive tumor focus should be done.
- Tumors that release cDNA into the peripheral blood at the primary tumor stage have a worse prognosis than those in which no cDNA is found in the blood. Consequently, in the future, liquid biopsy may play an increasingly important prognostic role in the differentiation of csDNA-positive and csDNA-negative tumors. This molecular staging can influence the decision for or against adjuvant therapy. The extent to which this concept is suitable for use in cancer practice remains to be seen; further clinical studies are needed.
- In general, due to methodological limitations, liquid biopsies, presumably, will never achieve the quality and information content of tumor tissue studies, but they complement them and have their own value in personalized molecular diagnostics and monitoring of tumor diseases. Another critical aspect is still the lack of standardization of technologies for the isolation and analysis of cDNA and quality control of sample processing.
Author Response
Dear reviewer,
Many thanks for making additional suggestions to further improve the paper.
Please find attached our response as well as the revised manuscript with tracked changes.
Maarten IJzerman
Reviewer 2:
The article is an original research that deserves the attention of the readers. I would like to note a few points that, in my opinion, should be emphasized:
1. Cell-free circulating tumor DNA is found in only about 75% of patients with metastatic tumor disease, with significant differences between different tumor forms and tumor stages. The concentration of cDNA is especially low in renal cell carcinomas, prostate and thyroid carcinomas, as well as in brain tumors (due to the blood-brain barrier), even with metastasis, so it is not always possible to diagnose it.
Re: we agree with this point and this was included in the original manuscript in line 425-429 with reference to mutational burden of the tumor. In addition, we now also include this in section 1.2, lines 108-111. We included one additional reference (Bettegowda et al, 2014).
2. Until now, there is no marker that could reliably distinguish ctDNA from tumor cells from cfDNA from healthy cells. Only the detection of a mutation known from the primary tumor gives confidence that the study worked, and the test result has clinical significance. However, if no known mutation is found in the liquid biopsy, then either there is no ctDNA in the blood, or the sensitivity of the assay is too low to detect existing cDNA. Also problematic in this context is the potential release of mutated DNA from benign, inflammatory layering lesions (eg, melanocytic skin nevus, intestinal polyp).
Re: we agree and have included a section discussing this following on the previous comment. This can be found in section 1.2, lines 115-116. We also refer to lines 425-429 (section 4.3) where this is discussed in the context of new methylation approaches.
3. Detection of ctDNA in blood does not allow drawing conclusions about where the tumor is located and which organ is affected. This should be further explored using imaging diagnostics.
Re: we have included this in the original manuscript in section 4.1, lines 390-394 discussing the benefits of a targeted screening / tumor-informed assays. In addition, we now have made reference to imaging as a further diagnostic tool to identify origin of the tumor in pan-cancer screening in lines 395-396.
We also added a comment to the abstract about imaging, see line 39.
4. Liquid biopsies have a high positive but low negative predictive value. For example, detection of an EGFR mutation from blood can be used as a prognostic marker to predict the response to targeted therapy with tyrosine kinase inhibitors, but in the absence of an EGFR mutation in the blood, the tissue sample must be further studied. Cases in which the EGFR mutation is not detected on liquid biopsy, but is detected on biopsy of tumor tissue are relatively common, and therefore, in the absence of signs of mutation in the blood, confirmation of its presence in tumor tissue should be sought.
Re: yes, we agree and have added this to section 4.1 (line 112-113). It follows from the first and second comment, that ctDNA may not be detected in metastatic cancers thereby inducing false-negative conclusions.
5. An interesting and promising field of application of liquid biopsy is the identification of emerging resistance mutations during tumor progression after therapy in order to adapt further treatment. Liquid biopsy plays an important role in finding the p.T790M resistance mutation in the EGFR gene in targeted therapy for non-small cell lung cancer. An additional molecular test for detecting this mutation of resistance to p.T790M-EGFR can be performed quickly and reliably based on ctDNA from blood plasma. However, the sensitivity of this method is only about 60-70%, therefore, in this situation, in the absence of evidence of mutation of resistance to EGFR, a second biopsy or sampling of cytological material from a progressive tumor focus should be done.
Re: we agree this is an interesting application and this is included in the manuscript (section 1.2, line 127-131). But we have not made any additional reference to the impact of the low sensitivity as we felt this was a bit too detailed. By including a section about sensitivity of detection and impact of diagnostic errors in section 1.2, we thought we had made this point sufficiently.
6. Tumors that release cDNA into the peripheral blood at the primary tumor stage have a worse prognosis than those in which no cDNA is found in the blood. Consequently, in the future, liquid biopsy may play an increasingly important prognostic role in the differentiation of csDNA-positive and csDNA-negative tumors. This molecular staging can influence the decision for or against adjuvant therapy. The extent to which this concept is suitable for use in cancer practice remains to be seen; further clinical studies are needed.
Re: we agree with this statement and this is also included in the manuscript while discussing the application in MRD and treatment selection following primary treatment. See also section 4.2, lines 410-418.
7. In general, due to methodological limitations, liquid biopsies, presumably, will never achieve the quality and information content of tumor tissue studies, but they complement them and have their own value in personalized molecular diagnostics and monitoring of tumor diseases. Another critical aspect is still the lack of standardization of technologies for the isolation and analysis of cDNA and quality control of sample processing.
Re: we agree regarding these additional comments but we think they are addressed in the manuscript. The issue of standardisation for isolation and analysis is added in section 4.2 (line 101 – 108), based on comments made by reviewer 1.
We also agree that liquid biopsies will not replace, rather complement tissue biopsies. But this also means we need to find the clinical utility and make a case to routinely translate liquid biopsies. Which patients, at what stage of disease and where in the clinical workflow. This is, to our opinion, also covered in the paper and in the conclusion. To be sure, we added a statement in the abstract of the paper (line 39).
